# Tailoring Electrochemical Performance of Co_3_O_4_ Electrode Materials by Mn Doping

**DOI:** 10.3390/molecules27217344

**Published:** 2022-10-28

**Authors:** Xingyu Liu, Mengdi Wang, Xiang Wu

**Affiliations:** School of Materials Science and Engineering, Shenyang University of Technology, Shenyang 110870, China

**Keywords:** Mn-doping, Co_3_O_4_, supercapacitor, energy density, specific capacitance

## Abstract

Reasonable design of electrode materials is the key to solving the low energy density of the supercapacitors. Transition metal oxide Co_3_O_4_ material is commonly used in the field of supercapacitors, but the poor cycle stability limits its practical application. Herein, we report 0.3Mn-Co_3_O_4_ nanostructures grown on nickel foam by a facile one-step hydrothermal approach. The morphology of the samples can be regulated by the introduction of different amounts of Mn ions. The specific capacitance reaches 525.5 C/g at 1 A/g. The performance of 0.3Mn-Co_3_O_4_ material is significantly improved due to its excellent stability and conductivity, which makes it a suitable electrode material for supercapacitors. A flexible asymmetric device is also fabricated using the sample as the cathode. The assembled capacitor still possesses a desirable cycle stability after charging and discharging of 10,000 times, and its capacitance retention rate can reach 83.71%.

## 1. Introduction

Many non-renewable energies have been excessively exploited and utilized in the past few years [1,2,3]. Therefore, it is imperative to develop clean energy with low pollution, low cost and sustainable utilization [4,5,6]. As an emerging energy storage system, the supercapacitor has attracted extensive attention due to its long cycle life, high power density, and fast charge and discharge process [7,8,9]. The structure of electrode materials is very important in determining the energy storage capacity of supercapacitors [10,11,12]. Among them, transition metal oxides are suitable alternatives because of their high specific capacitance [13,14,15,16,17]. 

Co_3_O_4_ material is considered to be a suitable candidate for supercapacitors due to its abundant reserves and high theoretical-specific capacitance [18,19]. In the literature, Gao et al. prepared Co_3_O_4_ products with a specific capacitance of 814 F/g, at a current density of 1 A/g [20]. A Co_3_O_4_ nanostructured composite with 473 F/g at 1 A/g is also synthesized through a simple hydrothermal reaction [21]. However, the Co_3_O_4_ electrode has low conductivity, leading to its poor cycle stability [22]. Some studies have shown that the conductivity of transition metal oxides can be improved by element doping. For example, Ojha et al. synthesized Mn-doped TiO_2_, and its capacitance and carrier density increased nearly twofold compared with the undoped samples [23]. Nguyen and co-workers report Mn-doped NiCo_2_O_4_ with a specific capacitance of 204.3 C/g at 1 A/g [24].

Herein, we synthesized a 0.3Mn-Co_3_O_4_ nanostructure by a simple one-step hydrothermal strategy. The specific capacitance of the sample reached 525.5 C/g at 1 A/g, and still maintained a high capacitance retention rate after 10,000 cycles. In addition, we assembled asymmetric supercapacitors (ASC) using the synthesized samples as the positive electrode and activated carbon as the negative one. The device delivers an energy density of 40.5 W/h kg at 1350 W/kg power density and a capacitance retention of 85.71% after 10,000 cycles.

## 2. Experimental Section 

In this work, all reagents were analytically pure and used without further purification. Before the synthesis, a piece of Ni foam (3 × 3 cm^2^) was immersed in 1.0 M HCl for 30 min. The Ni foam was then washed with deionized water and ethanol under ultrasound. Finally, the Ni foam was dried overnight at 60 °C.

### 2.1. Synthesis of 0.3Mn-Co_3_O_4_ Samples

2 mmol Co(NO_3_)_2_·6H_2_O, 10 mmol urea, and 4 mmol NH_4_F were successively placed in 60 mL deionized water for full agitation. After that, 0.3 mmol KMnO_4_ was added to the above solution under continuous stirring for 30 min. The homogeneous solution and the pretreated Ni foam were transferred into the reaction kettle at the same time. They were kept at 140 °C for 4 h. After the samples were cooled to room temperature, they were washed several times with deionized water and anhydrous ethanol, and dried overnight at 60 °C. Finally, the Co_3_O_4_ sample was obtained with a loading mass of 2.2 mg. Using the same steps as above, Mn-doped Co_3_O_4_ samples were obtained by adding 0.1, 0.3, and 0.5 mmol KMnO_4_ into the prepared solution, respectively. For the convenience of description, the above samples were named as 0.1Mn-Co_3_O_4_, 0.3Mn-Co_3_O_4_, and 0.5Mn-Co_3_O_4_. The mass loadings are 1.2, 1.1, and 1.5 mg, respectively.

### 2.2. Material Characterization

The crystal structure and element distribution of the samples were determined by an X-ray diffraction analyzer (XRD, Shimadzu-7000) and an X-ray photoelectron spectroscopy (XPS, ESCALAB250). The morphology of the samples was characterized by scanning electron microscopy (SEM, Gemini, 300-71-31).

### 2.3. Electrochemical Measurements

The electrochemical performance of the as-prepared samples was measured by electrochemical workstation (CHI660E) in a standard three-electrode system. Cyclic voltammetry (CV) curves, galvanostatic charge-discharge (GCD) curves, and electrochemical impedance spectroscopy (EIS) measurements were tested in 6 M KOH aqueous electrolyte. In the three-electrode system, the as-fabricated samples were used as the working electrode, Pt foil as the counter electrode, and Hg/HgO as the reference electrode. The specific capacitance (Cs) of the electrode materials was calculated by GCD curves using the follow equation:Cs = I∆t/m(1)
where Cs, I, ∆t and m are the specific capacitance, discharge current density (A/g), discharge time (s) and the mass of active materials (g), respectively. 

### 2.4. Assembly of Asymmetric Supercapacitors (ASC)

An asymmetric supercapacitor was assembled with a 0.3Mn-Co_3_O_4_ sample as the positive electrode, activated carbon as the negative electrode, and PVA-KOH gel as the electrolyte. The anode material was obtained by coating the Ni foam with a mixture of activated carbon (70 wt %), carbon black (20 wt %), and polyvinylidene fluoride (PVDF) (10 wt %) slurry, and dried at 80 °C for 24 h. Since the charge between the positive and negative electrodes is balanced, the mass ratio of cathode and anode can be calculated by using the following formula:q^+^ = q^−^(2)
q = cm∆V(3)
m^+^/m^−^ = C^−^V^−^/C^+^V^+^(4)

In the above formula, q is charge, C stands for specific capacitance, m represents the mass of active material, and V refers to the voltage window.

The corresponding energy density and power density of devices were calculated according to the equations: E = 1/2CV^2^(5)
P = 3600E/t(6)

E and P are energy density (Wh/kg) and power density (W/kg), respectively. 

## 3. Results and Discussion

The crystal structures of the prepared samples are characterized by X-ray diffraction spectroscopy (XRD). As can be observed in Figure 1a, two of the XRD patterns match the structure of Co_3_O_4_ (JCPDS No. 74-2120). The clear and sharp diffraction peaks at 2θ values of 18.99°, 31.27°, 36.84°, 44.81°, 59.35°, and 65.23°, can be indexed to the (111), (220), (311), (400), (511), and (440) crystal planes of Co_3_O_4_, respectively. By comparing the two XRD patterns, it can be proven that the addition of Mn source does not show a new diffraction peak appearing, which indicates that there is no new crystalline phase after doping [25]. In order to further study the structure of the samples, the surrounding area of the crystal plane (311) is enlarged, as depicted in Figure 1b. It can be seen that the diffraction peak of the 0.3Mn-Co_3_O_4_ sample is shifted to a low angle. This finding can be associated with Bragg’s diffraction formula, which is as follows:2dsinθ = nλ(7)
where n is an integer multiple of the wavelength, d is the distance between parallel atomic planes, λ is the incident wave length, and θ is the angle between the incident light and the crystal plane.

According to the Bragg diffraction formula, θ is inversely proportional to λ. The 0.3Mn-Co_3_O_4_ sample is shifted to a low angle, which proves that the incident wavelength of sample 0.3Mn-Co_3_O_4_ is increased compared with that of the Co_3_O_4_ sample. In general, the XRD patterns of the two samples shows that Mn is successfully incorporated into the Co_3_O_4_ lattice.

The structure of the prepared samples are further investigated by XPS. The XPS survey spectra of the 0.3Mn-Co_3_O_4_ sample are shown in Figure 1c. The peak of element Ni is primarily from nickel foam. Figure 1d depicts the XPS spectra of Co 2p. The spin-orbit peaks of Co 2p_3/2_ and Co 2p_1/2_ are located at 781.2 and 796.9 eV, respectively [26]. Moreover, the peaks with binding energies at 780.9 and 795.9 eV are attributed to Co^3+^ ions, while those with binding energies at 782.9 and 797.9 eV are attributed to Co^2+^ ions [27]. In addition, a pair of satellite peaks of the sample locate were at 786.7 eV and 804.1 eV. It can be seen that the peak of the Mn doped sample shows a negative shift of 0.4 V. This indicates that electron transfer occurs between Co_3_O_4_ and Mn atoms [28].

The XPS spectra of sample O 1s is shown in Figure 1e, which consists of three main peaks. Where O_def._ (532.8 eV) represents the oxygen defect state, O_surf._ (531.3 eV) shows the surface oxygen, and O_latt_ (529.3 eV) denotes the surface lattice oxygen [29]. Similarly, the spectra of the Mn doped Co_3_O_4_ sample is compared with the undoped sample. It demonstrates that the proportion of oxygen defects increases significantly when Mn is doped. This phenomenon is most likely due to the electrical imbalance caused by the substitution of Mn ions for Co ions during doping [30]. In order to maintain the electrical balance of the system, oxygen ions obtain electrons to form oxygen and leave the lattice, thus forming oxygen vacancies. The appearance of these oxygen vacancies increases the ion transport rate to some extent, exposing more electrochemical active sites. Figure 1f depicts the Mn 2p spectra of the 0.3Mn-Co_3_O_4_ sample. The peaks located at 639.1, 642.6 and 645.9 correspond to the Mn ions with different valence states: Mn^2+^, Mn^3+^, and Mn^4+^, respectively [31]. It is evident that the area occupied by tetravalent Mn ions is larger than that of divalent Mn ions, which makes the sample more stable.

Next, we observed the morphology of the as-prepared samples. The SEM images of the Co_3_O_4_ sample and the Mn-doped Co_3_O_4_ sample, at different magnifications, can be observed in Figure 2a–h. SEM images of the Co_3_O_4_ sample (Figure 2a,e) show that many needle-like nanowires grow uniformly on the Ni foam, and that they adhere with each other. With the addition of the Mn source, these nanowires are gradually connected to form sheets, as shown in Figure 2b, until they are completely replaced by nanosheets (Figure 2c). Due to the difference in the valence state and radius between the Mn ion and the Co ion in the original sample, the defects are inevitably introduced after Mn doping, leading to the occurrence of the above phenomena [32]. In Figure 2g, it can be observed that the nanosheet arrays grow vertically on the nickel foam. The gaps are formed between the nanosheets, allowing the active substance to come into full contact with the electrolyte. However, with the further increase of the Mn sources, it can be seen in Figure 2d that the gaps between the nanosheets become narrow, and a superposition between the nanosheets appears. To some extent, these phenomena reduce the electrochemical storage capacity of the samples.

Next, we studied the electrochemical performance of different samples in 6 M KOH. Figure 3a shows the electrochemical energy storage schematic of the 0.3Mn-Co_3_O_4_ nanosheet as electrode. Figure 3b shows the CV curves of the samples at 100 mV/s. It can be seen that the 0.3Mn-Co_3_O_4_ sample provided a larger area than the other samples, at the same scanning rate. In Figure 3c, the GCD curves of the samples are used to further study the capacitance behavior. At 1 A/g, the capacitances of Co_3_O_4_, 0.1Mn-Co_3_O_4_, 0.3Mn-Co_3_O_4_, and 0.5Mn-Co_3_O_4_ are 142.8, 433.6, 525.5, and 289.8 C/g, respectively. When the content of the Mn source reaches 0.3 mmol, the sample possesses the largest specific capacitance. The capacitance variation curves (Figure 3d) are consistence with the above results.

The EIS of the samples is shown in Figure 3e. The following parameters are used to test the EIS: a frequency range from 0.01 to 10,000 HZ, the perturbation amplitude is 0.01 V, and potential (0 V) during execution. It can be observed that a semi-circular arc appears in the lower left corner, where the high frequency region is actually the charge transfer resistance (R_ct_). In addition, the real axis intercept is equivalent to series resistance (R_s_). A diagonal line with an angle of about 45° is called the Warburg impedance. It is evident that 0.3Mn-Co_3_O_4_ possesses the lowest Rs value of 0.47 Ω. In order to explore whether the electrode materials possess pseudocapacitance behavior during the charging and discharging process, the following formula can be used for calculation [33]:I = av^b^(8)

In the above equation, I is the peak current, v is the scan rate, a and b are constants. When the value of b is 0.5, the electrode material shows diffusion control behavior. If the value of b is in the range of 0.5–1, the electrode material demonstrates the coexistence of surface control behavior and diffusion control behavior. The value of b is 1, which means that the electrode material behaves as surface control behavior. In Figure 3f, the b values of Co_3_O_4_ and 0.3Mn-Co_3_O_4_ samples are 0.683 and 0.725, respectively, which confirms that the charging and discharging process of the two electrode materials is pseudocapacitance dominated behavior. Figure 3g,h reveals the contribution ratio of surface capacitance and diffusion capacitance of the two samples. As the large sweep speed causes the ions to tend to “skim” on the electrode surface, there was no chance for diffusion. Therefore, the contribution of surface control gradually replaces the contribution of diffusion control, as the scanning speed continues to increase. The contribution rate of surface control and diffusion control can be evaluated by the following formula:i(v) = k_1_v + k_2_v^1/2^(9)

The values of k_1_ and k_2_ in the equation can be calculated by CV curves. In addition, the cyclic stability is also an important factor in assessing the electrochemical performance of the samples. In Figure 3i, the 0.3Mn-Co_3_O_4_ sample still maintains high capacitance after 10,000 times cycles, indicating its excellent cycle stability. To ensure the consistency of the experimental test results, the above CV curves, GCD curves, EIS curves, and cycle curves are all tested with three electrodes. In addition, the reference electrode and electrolyte are also consistent, which are Hg/HgO and 6 M KOH. In Table 1, the 0.3Mn-Co_3_O_4_ electrode is comparable with other previous reports [20,21,23,24,34].

To further explore the performance of the prepared material in practical application, we assembled a flexible asymmetric supercapacitor with the 0.3Mn-Co_3_O_4_ sample as the positive electrode and AC as the negative electrode. The schematic of an assembled device is illustrated in Figure 4a. Figure 4b shows the CV curve of the 0.3Mn-Co_3_O_4_ sample and the AC electrode. The curves prove that the anode and cathode materials match well, and the voltage of this device can be extended to 1.6 V. The assembled device is tested in a two-electrode system with KOH gel as the electrolyte. The CV curves of the device at different scanning speeds are shown in Figure 4c. The shapes of the CV curves do not change with the increase in scanning speed, but its area changes in direct proportion to the scanning speed. Figure 4d depicts the CV curves of the device at different voltage windows, at the scanning speed of 100 mV/s. The shapes of the CV curves remain stable as the area increases. The above phenomena indicate that the device can remain stable at 1.6 V voltage. The GCD curves of the device at different current densities are also shown in Figure 4e. When the current density is 0.5 A/g, the discharge time of the device can reach 116 s, and the specific capacitance is 116 C/g. Figure 4f shows the EIS of the device. The R_s_ value is 0.95 Ω, indicating that it possesses a low internal resistance. From the Ragone plot in Figure 4g, the 0.3Mn-Co_3_O_4_/AC//ASC achieves a maximum energy density of 43.5 W h/kg, at a power density of 1350 W/kg. The corresponding energy and power densities of these supercapacitors are shown in Table 2 [21,35,36,37,38]. To explore the cycling stability of the device, 10,000 charge and discharge tests were carried out. The device maintained a capacitance retention of 83.71% after testing (Figure 4g). The inset is an SEM image of the sample after cycling, which proves that the structure of the sample remains well preserved. 

## 4. Conclusions

In summary, we have prepared Mn ion doped Co_3_O_4_ samples by a simple one-step hydrothermal strategy. The morphology and crystal structure of the Co_3_O_4_ samples are regulated by changing the amount of Mn element. The addition of Mn changes the morphology of the sample from nanowires to nanosheets, and the thickness of the nanosheets also changes with the addition of Mn. The doping of Mn can regulate the surface morphology and oxygen vacancies. It greatly improves the conductivity of the electrode material and creates more active sites for electrochemical reaction. Moreover, the obtained samples maintain satisfactory energy density and power density after being assembled into devices. This demonstrates that the prepared materials possess great potential as a future energy storage device. 

## Figures and Tables

**Figure 1 molecules-27-07344-f001:**
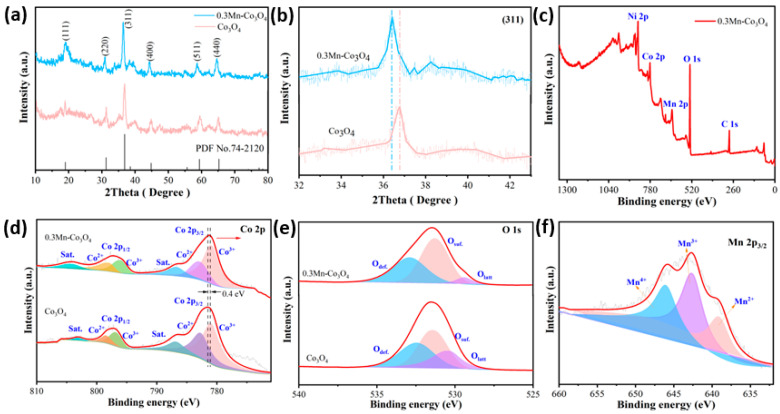
(**a**,**b**) XRD patterns of the as-prepared samples (**c**) the XPS survey spectra of 0.3Mn-Co_3_O_4_ sample (**d**) Co 2p (**e**) O 1s (**f**) Mn 2p.

**Figure 2 molecules-27-07344-f002:**
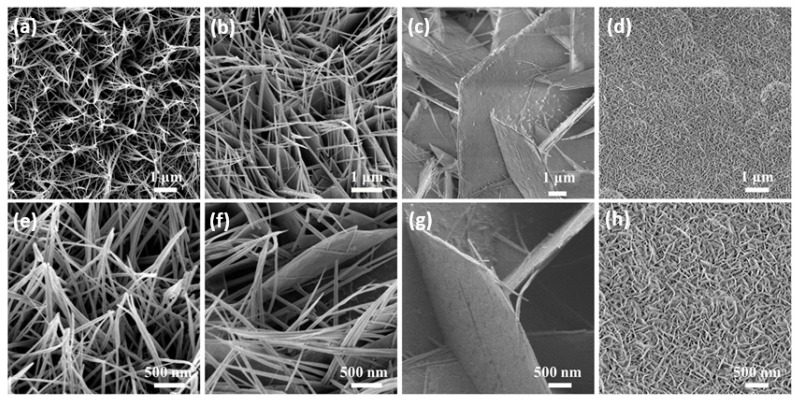
(**a**–**h**) Low and high magnification SEM images of the as-prepared samples (**a**,**e**) Co_3_O_4_ (**b**,**f**) 0.1Mn-Co_3_O_4_ (**c**,**g**) 0.3Mn-Co_3_O_4_ (**d**,**h**) 0.5Mn-Co_3_O_4._

**Figure 3 molecules-27-07344-f003:**
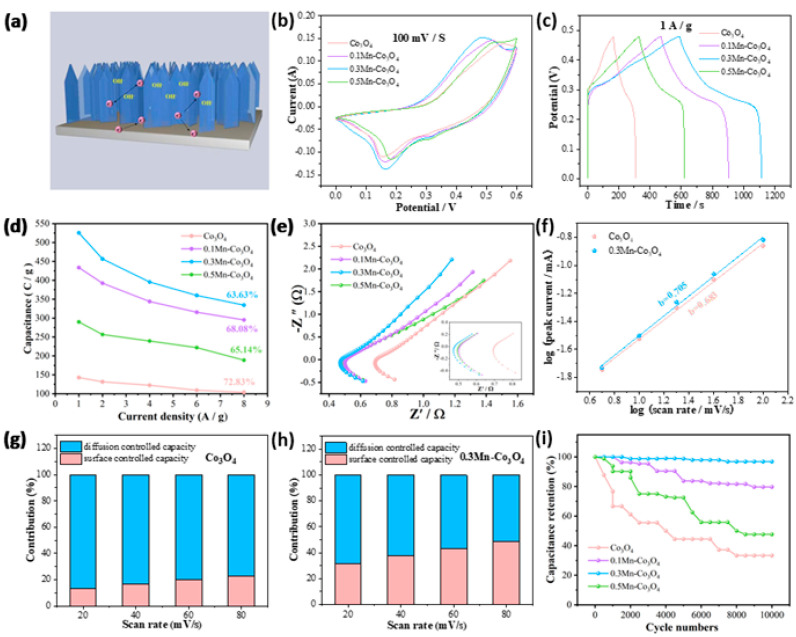
Electrochemical characterizations of three electrodes (**a**) The electrochemical mechanism of the 0.3Mn-Co_3_O_4_ nanosheet electrode (**b**) CV curves (**c**) GCD curves (**d**) Specific capacitance (**e**) Nyquist plots (**f**) b values of Co_3_O_4_ and 0.3Mn-Co_3_O_4_ (**g**,**h**) Contribution ratio between capacitance and diffusion-limited one (**i**) Cycling performance.

**Figure 4 molecules-27-07344-f004:**
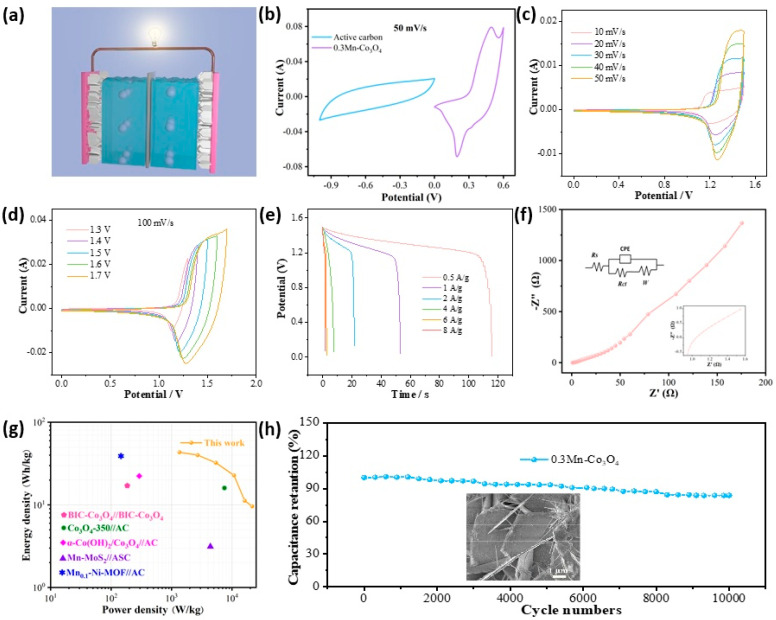
Electrochemical characterizations of ASC (**a**) Schematic (**b**) CV curves of 0.3Mn-Co_3_O_4_ and active carbon (**c**) CV curves with various scan rates (**d**) CV curves with various voltage windows (**e**) GCD curves (**f**) Nyquist plots (**g**) Ragone plot (**h**) Cycling performance.

**Table 1 molecules-27-07344-t001:** Electrochemical performances of several electrode materials.

Electrode Materials	Capacity	Current Density	Cycling Performance	Ref.
ZnO/Co_3_O_4_-450	567.5 (C/g)	1 (A/g)	83% after 5000 cycles	[20]
BIC-Co_3_O_4_	236.5 (C/g)	1 (A/g)	77% after 5000 cycles	[21]
1Mn-TiO_2_	164.4 (C/g)	1 (A/g)	84% after 2000 cycles	[25]
NMCO	204.3 (C/g)	1 (A/g)	99% after 1000 cycles	[26]
RGO/Co_3_O_4_ (94.3%)	198.6 (C/g)	5 (A/g)	122% after 5000 cycles	[31]
0.3Mn-Co_3_O_4_	525.5 (C/g)	1 (A/g)	96.83% after 10,000 cycles	This work

**Table 2 molecules-27-07344-t002:** Comparison of energy density and power density of 0.3Mn-Co_3_O_4_ device with previous reports.

Electrode Materials	Energy Density (W h/kg)	Power Density (W/kg)	Current Density	Ref.
BIC-Co_3_O_4_//BIC-Co_3_O_4_	17	184	0.1 (A/g)	[21]
Co_3_O_4_-350//AC	16.25	7500	10 (A/g)	[32]
α-Co(OH)_2_/Co_3_O_4_//AC	22.3	290	0.5 (A/g)	[33]
Mn-MoS_2_//ASC	3.14	4346.35	10 (A/g)	[34]
Mn_0.1_-Ni-MOF//AC	39.6	143.8	5 (mA/cm^2^)	[35]
0.3Mn-Co_3_O_4_/AC//ASC	43.5	1350	0.5 (A/g)	This work

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
