# Peer review of "Tailoring Electrochemical Performance of Co_3_O_4_ Electrode Materials by Mn Doping"

_molecules, 2022, doi:10.3390/molecules27217344_

Round 1

Reviewer 1 Report

This report (ID: molecules-1973844) is entitled “Tailoring electrochenical performance of Co3O4 electrode materials by Mn dopin” by research group Xiang Wu. This article is interesting and can make a good contribution to the scientific literature. However, the authors must address formal flaws and strengthen the presentation and clarity. The reviewer suggests a major revision is needed before publication in a peer-reviewed journal. Some specific comments on the manuscript:

1. ORIGINALITY:

The authors have to state the novelty of materials and list the advantages of these materials.

2. ABSTRACT:

The suggestion to the authors is to revise the abstract, summarize the problem, mention novelty, share significant results, and some conclusions. Suggest future implications.

3. METHODOLOGY:

*The authors suggest the possibility of extending these methodologies to develop practical applications.  

*Ensure this material has the potential to develop practical applications.

4. RESULTS AND DISCUSSION:

*Present results with scientific clarity and state the mechanisms.  

*Describe the cost-effectiveness and advantages of materials.

*improve flow, English. Results are briefly described and need comprehensive discussion.

* enhance results and discussion and focus on scientific aspects.

*If any, specify the limitations of developed materials at the end of the section.

5. CONCLUSIONS:

* reflect research results and compare with previous reports on this material, specify the improvements.

*Focus on the scope (such as future sustainability, cost-effectiveness, improvement in performance, and future implications)

Author Response

Reviewer 1

This report (ID: molecules-1973844) is entitled “Tailoring electrochenical performance of Co3O4 electrode materials by Mn dopin” by research group Xiang Wu. This article is interesting and can make a good contribution to the scientific literature. However, the authors must address formal flaws and strengthen the presentation and clarity. The reviewer suggests a major revision is needed before publication in a peer-reviewed journal. Some specific comments on the manuscript:

  1. ORIGINALITY:

The authors have to state the novelty of materials and list the advantages of these materials.

Response to comment: Thanks the reviewer’s comments. We added the related description as follow:

“...The synthesized 0.3Mn-Co3O4 material can greatly improve its electrochemical performance compared with the original Co3O4 material due to the low cost, high specific capacitance and good stability and conductivity, which makes it a suitable electrode material for supercapacitors.”

  1. ABSTRACT:

The suggestion to the authors is to revise the abstract, summarize the problem, mention novelty, share significant results, and some conclusions. Suggest future implications.

Response to comment: Thanks the reviewer’s comments. We have made changes to the abstract as described below:

“...The synthesized 0.3Mn-Co3O4 material can greatly improve its electrochemical performance compared with the original Co3O4 material due to the low cost, high specific capacitance and good stability and conductivity, which makes it a suitable electrode material for supercapacitors.”

 “...The device still possesses desirable cycle stability after 10000 times charging and discharging, and its capacitance retention rate can reach 83.71%.”

  1. METHODOLOGY:

*The authors suggest the possibility of extending these methodologies to develop practical applications.

Response to comment: Thanks the reviewer’s comments. We have added some experimental details in the revised version.

*Ensure this material has the potential to develop practical applications.

Response to comment: Thanks the reviewer’s comments. As mentioned in the answer to the previous question, we have added some specific descriptions to the details of the experiment, which increased the feasibility of the experimental method being repeated. In addition, the drugs used in the experiment are affordable and easily available, which has the potential to develop practical applications.

  1. RESULTS AND DISCUSSION:

*Present results with scientific clarity and state the mechanisms.

Response to comment: Thanks the reviewer’s comments. In the third section of the manuscript, we not only add some details of the experimental operation, but also add a description of the method to evaluate the contribution rate of diffusion capacitance and surface capacitance. We hope the above changes can make the content of the article more clear and easy to understand.

*Describe the cost-effectiveness and advantages of materials.

Response to comment: Thanks the reviewer’s comments. Both the cost-effectiveness and the advantages of this material are mentioned in the abstract and the conclusion, which are described as follows:

“...The synthesized 0.3Mn-Co3O4 material can greatly improve its electrochemical performance compared with the original Co3O4 material due to the low cost, high specific capacitance and good stability and conductivity, which makes it a suitable electrode material for supercapacitors.”

 “   In addition, the cost of materials used in this study is reasonable and the experimental process is simple, which is of great practicability.”

*improve flow, English. Results are briefly described and need comprehensive discussion.

Response to comment: Thanks the reviewer’s comments. We have added a lot of descriptions and details to the third chapter of the manuscript to make it more easier to understand.

* enhance results and discussion and focus on scientific aspects.

Response to comment: Thanks the reviewer’s comments.  We have given more discussion.

*If any, specify the limitations of developed materials at the end of the section.

Response to comment: Thanks the reviewer’s comments. We add the following reflections on limitations at the end of this section of the manuscript:

“   All the above results indicate that this material has potential and prospect as an electrode material for supercapacitors, but it still needs to be further explored to be applied in practice.”

  1. CONCLUSIONS:

* reflect research results and compare with previous reports on this material, specify the improvements.

Response to comment: Thanks the reviewer’s comments. In the conclusion, combined with the work done in this research, we have added the following related elaboration:

“   The addition of Mn makes the original morphology of the sample from nanowires to nanosheets, and the thickness of the nanosheets also changes with the addition of Mn.”

*Focus on the scope (such as future sustainability, cost-effectiveness, improvement in performance, and future implications)

Response to comment: Thank the reviewer for this valuable suggestion. In the revised version, we have made changes based on this suggestion, and added the relevant description:

“    In this study, the transition metal oxides are modified by doping ions, which greatly promotes the application of Co3O4 materials in the electrode materials of supercapacitors. The study also provides some reference for synthesis of other transition metal oxides.”

Reviewer 2 Report

The manuscript shows a good level a characterization and electrochemical analysis. The general quality of the manuscript is good, in terms of writing and conclusions made over the performed measurements. The level of originality is low, as the material has been extensively used. The novelty relies in the simplicity of the synthesis, as claimed by the authors. The relevance to the field is average, as a particular combination was obtained as best from the studied conditions but lacking from fundamental indications deriving in this particular formula/stoichiometry.  However, there are several considerations and questions prior to be considered for publication. The questions/recommendations are enlisted as follows:

1.      In the Abstract section, mention the specific values obtained from the stability tests (retained capacity after specific number of cycles).

2.      Check grammar and typos in general.

3.      In section 3, there is some amorphicity in the diffractograms, and differences are found at 188.99° (2-thetha), this has to be considered for the argumentation in the material differences. Maybe no new crystalline phases are found, but new planes are appearing. Explain further this point.

4.      In the XPS part, no information about the deconvolution process is given. Also how were the peak positions treated? Were they fixed or given certain flexibility and how much? This is important as a fitting process attempts to understand and explain how the components in the formation of the several species play a role. All this information is required and suggested to be synthesized in a table (Supplementary information, perhaps).

5.      Electrochemical measurements should be correctly reported, as the X-axis legends do not indicate if they are using a 3- or 2-electrode cell configuration. The potential axis should indicate which reference electrode (if one was used) was used. Always mention the electrolyte used.

6.      What are the reactions taking place at 0.25 and 0.45 V in Figure 3b??

7.      EIS part should be better described. Please use a correct representation of the EIS data that is, same scale for one unit in X-axis should be the same as Y-axis (square axes). Was an equivalent circuit used to fit this data. If not, experimental measurements should be represented as points not linked by a line, since it is understood commonly that the line corresponds to the fitted results. If fitted to a equivalent circuit, then present it and the data obtained from the fitting. Also, report the parameters used in EIS (frequency range, perturbation amplitude, number of measurements taken for the averaging, potential at which it was performed, etc).

8.      Please, correctly reference equation 8. There are assumptions that are not shown where they came from (b values).

9.      What procedure was used to evaluate the surface capacitance contribution from the diffusion capacitance? Include it in the body of the text.

10.   There seems to be a problem with scheme in Figure 4a. In the text is described as asymmetric, but in the figure looks symmetrical.

11.   In Figure 4c, there is a small difference at the 10 mV/s scan. Please explain this difference. What could it be attributed to?

12.   Figure 4.e can be represented as Fig 3.d.

13.   Elaborate deeper conclusions and link results, as several measurements were performed.

Author Response

please read the attached

Round 2

Reviewer 1 Report

Accept